# Meta-Analysis of Oxidative Transcriptomes in Insects

**DOI:** 10.3390/antiox10030345

**Published:** 2021-02-25

**Authors:** Hidemasa Bono

**Affiliations:** 1Program of Biomedical Science, Graduate School of Integrated Sciences for Life, Hiroshima University, 3-10-23 Kagamiyama, Higashi-Hiroshima, Hiroshima 739-0046, Japan; bonohu@hiroshima-u.ac.jp; Tel.: +81-82-424-4013; 2Database Center for Life Science (DBCLS), Joint Support-Center for Data Science Research, Research Organization of Information and Systems, 178-4-4 Wakashiba, Kashiwa, Chiba 277-0871, Japan

**Keywords:** insect, meta-analysis, oxidative stress, public database, RNA-seq, transcriptome

## Abstract

Data accumulation in public databases has resulted in extensive use of meta-analysis, a statistical analysis that combines the results of multiple studies. Oxidative stress occurs when there is an imbalance between free radical activity and antioxidant activity, which can be studied in insects by transcriptome analysis. This study aimed to apply a meta-analysis approach to evaluate insect oxidative transcriptomes using publicly available data. We collected oxidative stress response-related RNA sequencing (RNA-seq) data for a wide variety of insect species, mainly from public gene expression databases, by manual curation. Only RNA-seq data of *Drosophila melanogaster* were found and were systematically analyzed using a newly developed RNA-seq analysis workflow for species without a reference genome sequence. The results were evaluated by two metric methods to construct a reference dataset for oxidative stress response studies. Many genes were found to be downregulated under oxidative stress and related to organ system process (GO:0003008) and adherens junction organization (GO:0034332) by gene enrichment analysis. A cross-species analysis was also performed. RNA-seq data of *Caenorhabditis elegans* were curated, since no RNA-seq data of insect species are currently available in public databases. This method, including the workflow developed, represents a powerful tool for deciphering conserved networks in oxidative stress response.

## 1. Introduction

With increasing data collection, a statistical analysis called meta-analysis, which combines the results from multiple studies, is being widely used. By the end of 2020, over 200,000 records could be retrieved from PubMed by searching the keyword “meta-analysis”. This non-typical data analysis especially designed for the collected data is now more often employed, with scientists becoming more aware of its potential and usefulness, paving the way for new findings that could not be achieved with standard hypothesis-driven research methods [1]. Despite its useful analytical power, performing a meta-analysis can also be challenging owing to difficulties in collecting data from public databases and analyzing them. Nonetheless, this approach has been often used for investigating human-related subjects, allowing us to overcome major setbacks in human research, such as limited primary sample collection and small patient cohorts.

In the life sciences, several datasets have been archived in public databases that can be freely accessed and reused; this is particularly true for nucleotide sequence data. The database for nucleotide sequence, the Sequence Read Archive (SRA), now contains over 20 Petabases of open access nucleotide sequences (https://www.ncbi.nlm.nih.gov/sra/docs/sragrowth/ (accessed on 1 February 2021)). For example, chromatin immunoprecipitation sequencing (ChIP-seq) data in the SRA database were collected, curated, and pre-calculated for re-use in the ChIP-Atlas database [2]. An increasing number of meta-analyses can be found that re-use the public sequence data on a large scale, especially for coronavirus disease 2019 (COVID-19) studies. A tool for gene set enrichment analysis re-using biomedical data for COVID-19, named Coronascape, makes it possible to analyze the functional features of a gene list, specifically for COVID-19 studies [3]. However, few studies re-use those resources when considering the amount of available data. In particular, transcriptomes exposed to stress are a very interesting target for data meta-analysis. For example, hypoxic transcriptomes could be collected from public gene expression data from human and mouse cell lines and integrated with ChIP-seq data of transcription factors related to hypoxic stress by using the ChIP-Atlas database [4].

Reactive oxygen species (ROS), including peroxides, superoxides, and hydroxyl radicals, are highly reactive chemical molecules formed due to the electron acceptability of oxygen. ROS are generated as natural byproducts of energy metabolism and have crucial roles in cell signaling and homeostasis. Oxidative stress is a state of imbalance between the production of ROS and the ability of the biological system to directly detoxify ROS or repair the damage caused by them. When the normal redox state of a biological system is disturbed, peroxides and free radicals can damage proteins, lipids, and DNA, thereby resulting in the impaired function of various intercellular mechanisms/organelles. In humans, oxidative stress is believed to be involved in the development of cancer and Parkinson’s disease, among other pathological conditions [5]. We have explored the potential of silkworm (*Bombyx mori*) as a new model organism for investigating Parkinson’s disease [6] and superoxide dismutases [7,8]. Therefore, we focused on various effects of oxidative stress on *B. mori* instead of using cultured cells.

Based on the abovementioned information, the present study aimed to apply a meta-analysis approach to shed light on the oxidative transcriptomes of insects based on publicly available data. We first collected oxidative stress response-related RNA sequencing (RNA-seq) data from a wide variety of insect species. The collected gene expression data were manually curated and then systematically analyzed using a new RNA-seq workflow designed specifically for insects. The results were evaluated by two metric methods to construct a reference dataset for oxidative stress response studies. This study provides a method, including the workflow developed, that can be a powerful tool for deciphering conserved networks in oxidative stress response.

## 2. Materials and Methods 

### 2.1. Curation of Public Gene Expression Data

Several public databases for gene expression data are available, whereas only one repository archives nucleotide sequence data under the International Nucleotide Sequence Database Collaboration. This makes it difficult to re-use gene expression data available in public repositories. Therefore, we initially used the All Of gene Expression (AOE) online tool [9] to acquire oxidative stress-related gene expression data from public databases. AOE integrates metadata, not only from the Gene Expression Omnibus (GEO) database of the U.S. National Center for Biotechnology Information (NCBI) [10] but also from ArrayExpress of the European Bioinformatics Institute (EBI) [11] and Genomic Expression Archive (GEA) from the DNA Data Bank of Japan (DDBJ) [12]. Moreover, AOE covers RNA-seq data archived only in the SRA [13].

Conventional search using the keywords ‘oxidative’ and ‘insect’ yielded no hits because the metadata for those were not rich enough for the given query. Therefore, extensive searches using additional keywords (for example, ‘paraquat’ and ‘rotenone’ and others) were manually performed to populate the dataset. Detailed search in the NCBI GEO web interface was also used to assess records for insect species because some entries were not found by the AOE.

### 2.2. Retrieval and Quality Control of Sequence Data

For data retrieval from the SRA database and file conversion to FASTQ, we used the prefetch and fasterq-dump tools from the SRA Toolkit (v2.9.6) respectively [14]. A parallel implementation of gzip (pigz) was used to compress FASTQ files and reduce the time required for this process. For the trimming and quality control of reads from the SRA database, Trim Galore! (v0.6.6) [15] with Cutadapt (v1.18) [16] was applied to filter reads with unsatisfactory quality. This step is time-consuming, but it is indispensable for biologists to re-use reads archived in SRA, as the quality of reads is not uniform.

### 2.3. Gene Expression Quantification

Salmon was used for quantifying the expression of transcripts using RNA-seq data [17]. For the reproducibility of data analysis, Common Workflow Language (CWL) was introduced [18]. In the shell scripts to run Salmon, we deployed the CWL definition files of Salmon (indexing and quantification of single-end and paired-end reads) in Pitagora Workflows (v0.1.0) maintained by Pitagora Network, also known as Galaxy Community Japan [19]. Computer programs to run gene expression quantification, including retrieval and quality control of sequence data, were freely accessible in the GitHub repository [20].

The ratio of all gene pairs (termed ON-ratio) was calculated by pairing oxidative stress and normal states Equation (1). A small number (in this case, 0.01) was added for avoiding the calculation of logarithm of zero.
ON-ratio = log(gene expression_oxidative stress_ + 0.01) − log(gene expression_normal state_ + 0.01)(1)

ON-ratio values for all paired samples can be classified into three groups: upregulated, downregulated, or unchanged. When the ON-ratio was over the threshold for upregulation, the gene was treated as “upregulated.” Similarly, when the ON-ratio was under the threshold for downregulation, the gene was treated as “downregulated.” If the gene was treated as neither “upregulated” nor “downregulated,” it was classified as “unchanged.” The numbers of counts for upregulated, downregulated, and unchanged were calculated for all genes. Several thresholds were tested to optimize the calibration, and we adopted a 10-fold threshold for upregulation and 0.1-fold threshold for downregulation.

An oxidative stress-normal state score (ON-score) was then calculated, as shown in Equation (2). A detailed description of the ON-score is provided in the Result section.
ON-score = count number_upregulated_ − count number_downregulated_(2)

ON-ratio and ON-score were formally introduced as HN-ratio and HN-score in the meta-analysis of the hypoxic transcriptomes [4].

### 2.4. Functional Annotation and Gene Set Enrichment Analysis

Ensembl Biomart as used to extract the list of transcript stable identifiers (IDs) and gene names and corresponding Gene Ontology annotations for *Drosophila melanogaster* and *Caenorhabditis elegans* from the Ensembl database (v101) [21]. The orthologous gene relationship between these two species was also retrieved from Ensembl Biomart.

Metascape was used for gene set enrichment analysis [3]. In the enrichment analysis, the functional annotation provided by Metascape was used for the queried genes.

### 2.5. Visualization

In addition to joining two sets of data column by ID, we used TIBCO Spotfire Desktop (v7.6.0; TIBCO Software Inc., Palo Alto, CA, USA) to produce scatter plots and histograms.

## 3. Results

### 3.1. Data Collection

The overall procedure of our study is depicted in Figure 1. We first pursued oxidative stress-related transcriptome data from insect species in public databases using the AOE tool. Because the conventional keyword search by ‘oxidative’ and ‘insect’ in AOE yielded no hits, we manually curated the search results for ‘oxidative’ by limiting the species to insects. Most data only contained the word ‘oxidative’ in corresponding metadata without gene expression results for oxidative stress. Although several datasets in the public databases were quantified by microarrays, we limited the transcriptome data to RNA-seq data for the comparative study in the meta-analysis.

Currently, only RNA-seq data for *Drosophila melanogaster* can be found when using ‘insects’ as a keyword in the public databases. Therefore, we also explored potential data based on names of oxidative stress-related reagents in the NCBI GEO web interface. After curating all candidate data, pair sets (oxidative stress and control) were defined (Table 1).

### 3.2. Quantification of Transcripts Using RNA-seq

For analyzing human and mouse RNA-seq data, there exists a pipeline centered on Salmon called ikra [22]. Ikra not only automates the RNA-seq data analysis process but also makes it easy to re-use data from public databases for meta-analysis. Currently, no such pipeline exists for insect species. Therefore, we created a new generic pipeline for quantifying RNA-seq data called Systematic Analysis for Quantification of Expression (SAQE), which consists of the following steps:Retrieval of RNA-seq reads from the SRA databaseConversion of data format and compressionTrimming and quality control of readsTranscriptome assembly by TrinityExpression quantification by Salmon

Step 4 (transcriptome assembly) was not used in this study because well-curated transcriptome sequences were available for the target organisms. Nonetheless, this step is important for insects whose genome sequences are not available for data analysis.

For reproducibility of the study, Docker containers with fixed versions were employed in the last two steps. Furthermore, a script language, CWL, was partly employed in SAQE to call Docker containers, which is an open standard for describing analysis workflows and tools in scientific computing. Using CWL, one can replicate the analysis without installation of software, regardless of the type and version of operation system used (MacOS, Linux, and Windows). CWL enables the use of the workflow very easily, without installing Trinity, which users often find difficult to install on their computers. All codes of SAQE are freely available from GitHub [20].

As an output, SAQE yields the matrix for quantified expression data in transcript per million (TPM). This expression matrix was uploaded to figshare and is publicly available [23].

Using the expression matrix, hierarchical clustering for sample direction was performed. A dendrogram revealed that RNA-seq data from the same project clustered together (Appendix A), suggesting the need for analysis of differences in gene expression profile between samples subjected to oxidative stress and those in the normal state.

### 3.3. Differentially Expressed Genes under Oxidative Stress

We tried to evaluate differentially expressed genes under oxidative stress. By pairing oxidative stress and normal state samples, we calculated the ratio of all gene pairs (termed ON-ratio). The ON-ratio was previously introduced as HN-ratio (Hypoxia vs. Normoxia) in meta-analysis of hypoxic transcriptomes [4]. After detailed investigation of data, we employed in this study a smaller value (0.01, previously 1) for genes with weak expression. ON-ratio was calculated for all sample pairs (Table 1). The complete list of ON-ratio is publicly available [24].

For meta-analysis of these values, the simplest way is to use ‘averaged’ ON-ratio for all samples. The histogram of the averaged ON-ratio showed a skewed distribution curve (Figure 2A), with few genes having high ON-ratios and more genes with negative ON-ratios.

To evaluate oxidative stress-inducible genes, an oxidative stress-normal state score (ON-score) was calculated for all genes, as described in the Materials and Methods. The complete list of ON-score is also publicly available [25]. In contrast to ON-ratio, ON-score is quantal because it is calculated by subtracting the number of counts of upregulated from that of downregulated genes (Figure 2B). ON-score also showed skewed distribution curve, with more genes having ON-scores below zero. These results showed that several genes were downregulated under oxidative stress.

We then performed gene set enrichment analysis using Metascape [3]. The analysis revealed that the genes upregulated in many samples with high ON-scores were related to the cellular component morphogenesis (GO:0032989) and blood circulation (GO: 0008015) (Figure 3A). Metascape also uncovered the functions of downregulated genes with low ON-scores (Figure 3B), being related to organ system process (GO:0003008) and adherens junction organization (GO:0034332).

We then investigated ON-scores for genes annotated with GO:0006979 (response to oxidative stress), since the initial analyses did not show functional enrichment for oxidative stress-related annotated genes. In addition to GO:0006979, child terms of this GO group were included in the new differentially expressed gene analysis (Table 2).

Among the identified oxidative stress-related genes, many more genes had negative ON-scores (Figure 4).

### 3.4. Cross-Species Analysis of Transcriptomes under Oxidative Stress

For the cross-species study, we collected RNA-seq data from *C. elegans* because no RNA-seq data of insect species can be currently found in public databases. Similar to the approach used before, transcriptome sequencing data were curated for *C. elegans* and RNA-seq data were manually collected from the public databases (Table 3). In this dataset for *C. elegans*, rotenone was the most dominant reagent to cause oxidative stress, whereas paraquat was the most dominant reagent in the dataset for *D. melanogaster*.

The same procedure for retrieval and quality control of sequence data and the same gene expression quantification using SAQE as described above were used for this dataset [27]. Using calculated differential values (ON-ratio) [28] and ON-score [29], transcriptomes with oxidative stresses in *D. melanogaster* and *C. elegans* were compared and visualized in scatter plots (Figure 5). The integrated lists for ON-ratio and ON-score are publicly available [30,31]. There was no striking correlation observed between these two organisms. Nonetheless, genes with correlated expression pattern could be useful for candidate gene selection.

We next looked into genes annotated with GO:0006979 (response to oxidative stress). Upregulated experiments were dominant (ON-score positive) in nine genes (*CanA1*, *GstS1*, *Hsp22*, *Pde8*, *Sod2*, *foxo*, *per*, *ple*, and *rl*) in both *D. melanogaster* and *C. elegans*, whereas downregulated experiments were dominant (ON-score negative) in 13 genes (*CG9314, CG9416, CYLD, Ddc, Itp-r83A, ND-B17.2, Trap1, alph, bsk, mtd, park, ple,* and *whd*) (Table 4). More genes had distinct ON-scores but are not shown in Figure 5 because no *C. elegans* orthologs for those genes could be assigned.

The complete list of GO:0006979 (response to oxidative stress) annotated genes with ON-scores in Table 4 is publicly available [32].

## 4. Discussion

In the present study, we collected transcriptome data specific to oxidative stress responses from public databases, such as the NCBI GEO, EBI ArrayExpress, DDBJ GEA, and the SRA, using the AOE web tool. Although RNA-seq is now widely used for transcriptomic studies in a wide variety of insect species, RNA-seq data specific for oxidative stress conditions (with a normal control) were only found for *D. melanogaster,* in spite of extensive manual curation. In most cases, the dataset only contained oxidative stress RNA-seq data specific for knockdown (or knockout) conditions. Hence, continuous efforts should be made to collect data from public databases.

Collecting data is a critical and laborious step because there are many “oxidative” descriptions in the metadata of public databases. Moreover, many of these are “noise,” which do not contain any transcriptome data for oxidative stress responses. To overcome this handicap, we must re-organize public databases for semantic queries. Currently, there is no computational method to filter these ‘noise’ events, and manual curation by experts with domain knowledge is indispensable. The lack of information in databases can be addressed by searching published manuscripts (even supplemental materials in some cases). Deciphering metadata to collect the needed information from English texts also requires professional knowledge on sample preparation. Though a major hurdle, manual curation and creation of datasets are crucial for conducting this type of meta-analysis.

As a new generic pipeline for quantifying RNA-seq data, SAQE was developed for insect species without reference genome sequences [20]. Herein, SAQE was applied for insect species with reference transcriptomes in the public database. Only the transcriptome sequences of *D. melanogaster* and *C. elegans* were used, since both organisms are typical experimental models and their transcriptome sequence sets are well-curated. Moreover, as it uses transcriptome assembly obtained from the execution of Trinity, SAQE can also be applied for non-model organisms without reference transcriptomes. Technically, SAQE is not pure ‘workflow’ in CWL currently. It is a group of shell scripts making full use of CWL command line tools and workflows for practical use in insect study. Continuous development of SAQE itself is also needed.

The gene set enrichment analysis by Metascape for upregulated and downregulated genes showed interesting functional enrichments but is difficult to be interpreted in the context of oxidative stress. These results contrasted with those from the analysis for hypoxic transcriptomes in human cultured cells, which identified HIF1 TFPATHWAY, GO:0001666 response to hypoxia, and GO:0005996 monosaccharide metabolic process as being enriched. This is due to the various sources of oxidative stress and transcriptomic responses to oxidative stress, which are also not yet well-studied. Additionally, in insect species, functional annotation of genes is still underdeveloped, as compared with that in humans and mice. In other words, there are chances of making important discoveries from meta-analysis of these data.

Herein, the collected datasets were relatively small and biased. Only 19 pairs of oxidative stress and normal condition samples were collected compared with over hundred pairs of hypoxic and normoxic transcriptomes meta-analyzed in a previous study. In addition, the most represented stress source was paraquat (Table 1). Hence, more data and more species are needed for a better meta-analysis. Nevertheless, interesting genes could be found in this meta-analysis. Knockdown of *Sod2* in *Tribolium castaneum* was reported to impair its sensitivity to paraquat [33], and *Sod2* was one of the genes with positive ON-scores in both *D. melanogaster* and *C. elegans* (Table 4). Therefore, we believe that this meta-analysis provides a repository for future studies.

Our future work will gather oxidative stress-related transcriptome data for insect species to continuously update this meta-analysis. This approach will potentially uncover conserved mechanisms at the molecular level.

## 5. Conclusions

In this study, we tried to collect oxidative stress-related transcriptomic data in a wide variety of insect species from public databases. To date, such transcriptomes are only available for *D. melanogaster*; however, around 20 pairs of oxidative stress and normal state transcriptomes were listed. For analyzing such transcriptomes, an RNA-seq analysis workflow for species without reference genome sequences was developed. No distinct features were retrieved from enrichment analyses of upregulated and downregulated gene sets, but enrichments for other functional groups provided interesting new insights. We focused on genes annotated with oxidative stress-related Gene Ontology terms in the comparative analyses of *C. elegans* oxidative stress transcriptomes using the same curation process as that for *D. melanogaster*. This method, including the workflow developed, represents a powerful tool for deciphering conserved networks in oxidative stress response.

## Figures and Tables

**Figure 1 antioxidants-10-00345-f001:**
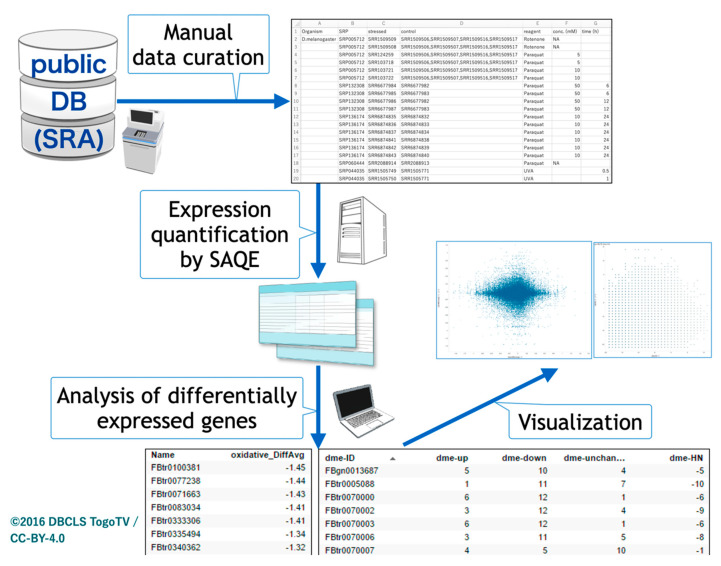
Schematic view of insect oxidative transcriptome meta-analysis. Public databases were searched, oxidative stress-related RNA sequencing data were manually curated, and a meta-analysis was performed.

**Figure 2 antioxidants-10-00345-f002:**
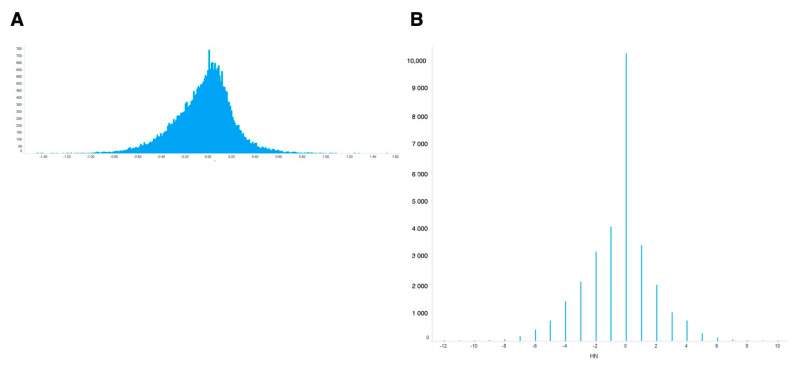
Distribution of calculated differential values for all genes. (**A**) Differential values averaged for all experiments. (**B**) Differential values normalized by newly introduced metrics (ON-score).

**Figure 3 antioxidants-10-00345-f003:**
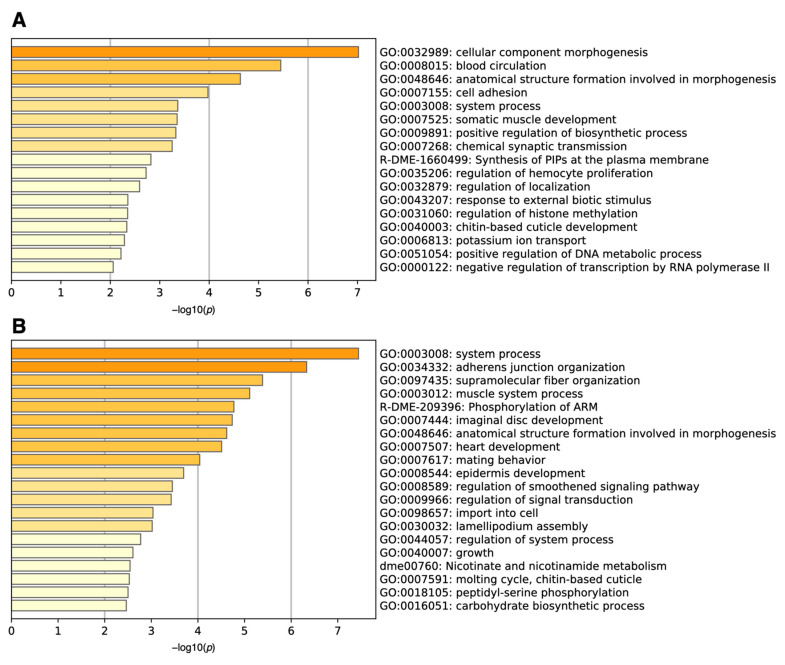
Histogram of gene set enrichment analysis for (**A**) upregulated (ON-score > 6) and (**B**) downregulated (ON-score < −7) genes under oxidative stress.

**Figure 4 antioxidants-10-00345-f004:**
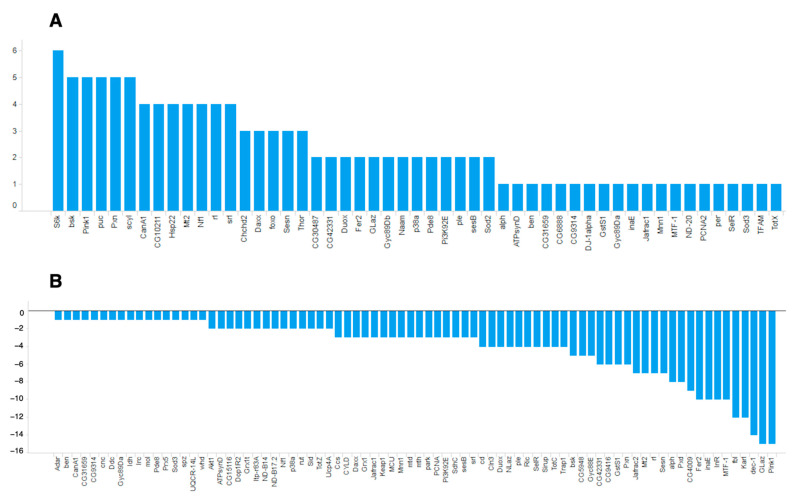
ON-scores for (**A**) upregulated and (**B**) downregulated genes annotated with GO:0006979 (response to oxidative stress).

**Figure 5 antioxidants-10-00345-f005:**
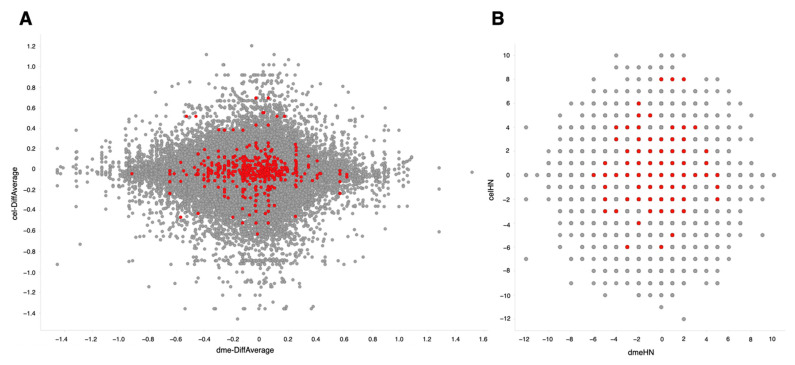
Comparison of differentially expressed genes between *Drosophila melanogaster* (X-axis) and *Caenorhabditis elegans* (Y-axis). Colored dots correspond to genes annotated with GO:0006979 (response to oxidative stress). (**A**) Differential values averaged for all experiments. (**B**) ON-score values.

**Table 1 antioxidants-10-00345-t001:** Dataset generated by the curation

SRA Project ID	Oxidative Stress	Control	Source of Stress (Reagent)	Conc. (mM)	Time (h)
SRP005712	SRR1509509	SRR1509506 ^1^	Rotenone	NA	NA
SRP005712	SRR1509508	SRR1509506 ^1^	Rotenone	NA	NA
SRP005712	SRR124259	SRR1509506 ^1^	Paraquat	5	NA
SRP005712	SRR103718	SRR1509506 ^1^	Paraquat	5	NA
SRP005712	SRR103721	SRR1509506 ^1^	Paraquat	10	NA
SRP005712	SRR103722	SRR1509506 ^1^	Paraquat	10	NA
SRP132308	SRR6677984	SRR6677982	Paraquat	50	6
SRP132308	SRR6677985	SRR6677983	Paraquat	50	6
SRP132308	SRR6677986	SRR6677982	Paraquat	50	12
SRP132308	SRR6677987	SRR6677983	Paraquat	50	12
SRP136174	SRR6874835	SRR6874832	Paraquat	10	24
SRP136174	SRR6874836	SRR6874833	Paraquat	10	24
SRP136174	SRR6874837	SRR6874834	Paraquat	10	24
SRP136174	SRR6874841	SRR6874838	Paraquat	10	24
SRP136174	SRR6874842	SRR6874839	Paraquat	10	24
SRP136174	SRR6874843	SRR6874840	Paraquat	10	24
SRP060444	SRR2088914	SRR2088913	Paraquat	NA	NA
SRP044035	SRR1505749	SRR1505771	UVA	NA	0.5
SRP044035	SRR1505750	SRR1505771	UVA	NA	1

^1^ SRR1509506, SRR1509507, SRR1509516, SRR1509517. Abbreviations: NA, not available; UVA, ultraviolet A radiation.

**Table 2 antioxidants-10-00345-t002:** Child terms (direct descendants) of the Gene Ontology term GO:0006979 (response to oxidative stress).

Child GO ID	Child GO Term	Relationship to GO:0006979
GO:0001306	age-dependent response to oxidative stress	is_a
GO:0033194	response to hydroperoxide	is_a
GO:1902882	regulation of response to oxidative stress	regulates
GO:1902883	negative regulation of response to oxidative stress	negatively_regulates
GO:0070994	detection of oxidative stress	is_a
GO:1902884	positive regulation of response to oxidative stress	positively_regulates
GO:0000302	response to reactive oxygen species	is_a
GO:0080183	response to photooxidative stress	is_a
GO:2000815	regulation of mRNA stability involved in response to oxidative stress	part_of
GO:0034599	cellular response to oxidative stress	is_a

Original source: [26].

**Table 3 antioxidants-10-00345-t003:** *Caenorhabditis elegans* oxidative transcriptome dataset generated by the curation.

SRA Project ID	Oxidative Stress	Control	Source of Stress (Reagent)	Condition
SRP070204	SRR3173735 ^1^	SRR3173720 ^2^	Sodium arsenite	10 mM, 5 h
SRP021083	SRR827426	SRR827423	Rotenone	100 nM
SRP021083	SRR827427	SRR827424	Rotenone	100 nM
SRP021083	SRR827428	SRR827425	Rotenone	100 nM
SRP021083	SRR827432	SRR827429	Rotenone	100 nM
SRP021083	SRR827433	SRR827430	Rotenone	100 nM
SRP021083	SRR827434	SRR827431	Rotenone	100 nM
SRP021083	SRR827438	SRR827435	Rotenone	100 nM
SRP021083	SRR827439	SRR827436	Rotenone	100 nM
SRP021083	SRR827440	SRR827437	Rotenone	100 nM
SRP021083	SRR827443	SRR827441	Rotenone	100 nM
SRP021083	SRR827444	SRR827442	Rotenone	100 nM
ERP117708	ERR3580218	ERR3580215	Gamma radiation	10 Gray
ERP117708	ERR3580219	ERR3580216	Gamma radiation	10 Gray
ERP117708	ERR3580220	ERR3580217	Gamma radiation	10 Gray
ERP117708	ERR3580221	ERR3580215	Gamma radiation	100 Gray
ERP117708	ERR3580222	ERR3580216	Gamma radiation	100 Gray
ERP117708	ERR3580223	ERR3580217	Gamma radiation	100 Gray
ERP117708	ERR3580224	ERR3580215	Gamma radiation	0.4 Gray
ERP117708	ERR3580225	ERR3580216	Gamma radiation	0.4 Gray
ERP117708	ERR3580226	ERR3580217	Gamma radiation	0.4 Gray

^1^ SRR3173735—SRR3173746. ^2^ SRR3173720—SRR3173734.

**Table 4 antioxidants-10-00345-t004:** *Drosophila melanogaster* oxidative stress-related genes with ON-score feature (positive values (+) or not (−)) and *Caenorhabditis elegans* ortholog (yes or no).

ON-Score	*C. elegans* Ortholog	Genes
+	yes	*CanA1, GstS1, Hsp22, Pde8, Sod2, foxo, per, ple, rl*
+	no	*CG10211, CG30487, CG31659, CG42331, CG6888, Chchd2, Daxx, GLaz, Gyc89Da, Gyc89Db, Jafrac1, MTF-1, Mnn1, Mt2, Nf1, Pi3K92E, SelR, TFAM, Thor, TotX, inaE, p38a, puc, scyl, srl*
−	yes	*CG9314, CG9416, CYLD, Ddc, Itp-r83A, ND-B17.2, Trap1, alph, bsk, mtd, park, ple, whd*
−	no	*Akt1, CG31659, CG4009, CG42331, CG5948, Ccs, Daxx, GLaz, Gyc88E, Gyc89Da, InR, Irc, Jafrac1, Jafrac2, Karl, Keap1, MTF-1, Mnn1, Mt2, NLaz, Nf1, Pi3K92E, Prx5, Pxd, Ric, SelR, Sid, TotC, TotZ, cd, cnc, dec-1, fbl, inaE, mth, p38a, rut, spz, srl*

## Data Availability

The data presented in this study are openly available in figshare at https://doi.org/10.6084/m9.figshare.c.5271731 (accessed on 1 February 2021). Source codes to replicate the study are also freely available at GitHub (https://github.com/bonohu/SAQE (accessed on 1 February 2021)).

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
