# Peer review of "Meta-Analysis of Oxidative Transcriptomes in Insects"

_antioxidants, 2021, doi:10.3390/antiox10030345_

Round 1

Reviewer 1 Report

General comments:

Current version of the manuscript has been well organized and written well. However, before publication, there are couple of minor mistakes that could be improved.

Specific comments:

In the section of “Abstract”, two points I want to make:

  1. 1st sentence has to be rewrite since it is a bit confused.
  2. At the very beginning of abstract, I want the authors to add the biological significance of oxidative stress, and oxidative transcriptomes in insects. Without the introduction, it will be very difficult to follow this manuscript, and does not help enhance the citation of this paper after publication.

In the section of “Results”

  1. 3.1. Data collection: It seems that there are so many short paragraphs, and thus I recommend the authors to combine the short paragraphs (2 or 3 if possible).

In the section of “Conculsions”

  1. Please, combine the short paragraphs.

Author Response

In the section of Abstract”, two points I want to make:

> 1. 1st sentence has to be rewrite since it is a bit confused.

Response:

Thank you for your suggestion. We have revised the first part of the Abstract as follows.

> Data accumulation in public databases has resulted in extensive use of meta-analysis, a statistical analysis that combines the results of multiple studies.

> 2. At the very beginning of abstract, I want the authors to add the biological significance of oxidative stress, and oxidative transcriptomes in insects. Without the introduction, it will be very difficult to follow this manuscript, and does not help enhance the citation of this paper after publication.

Response:

Thank you for your suggestion. We have revised the relevant text as follows.

> Oxidative stress occurs when there is an imbalance between free radical activity and antioxidant activity, which can be studied in insects by transcriptome analysis.

>In the section of Results”

> 3.1. Data collection: It seems that there are so many short paragraphs, and thus I recommend the authors to combine the short paragraphs (2 or 3 if possible).

Response:

Thank you for your suggestion. I have combined short paragraphs, so that there are now only two paragraphs in this section.

> In the section of Conculsions”

> Please, combine the short paragraphs.

Response:

Thank you for your suggestion. We have combined the short paragraphs.

Reviewer 2 Report

Manuscript Title: Meta-Analysis of Oxidative Transcriptomes in Insects.

The manuscript can be accepted with the following comments. 

Comments:
1. Introduction section has to be improved by including the detailed explanation of reactive oxygen species (ROS)

2. At the end of the introduction section, the author has to explain aim, objective and outcome of the present study. 

3. What is the importance of Common Workflow Language? Why author had used this one in the present study? please justify.

4. Table 3 should be uniform, there was an alignment issues by end of the table.

5. In Table 4, Please give an expansion of "+" and "-"

Author Response

Reviewer 2

> 1. Introduction section has to be improved by including the detailed explanation of reactive oxygen species (ROS)

Response:

Thank you for your suggestion. We have added a detailed description about reactive oxygen species (ROS) in the Introduction as follows.

> Reactive oxygen species (ROS), including peroxides, superoxides, and hydroxyl radicals, are highly reactive chemical molecules formed due to the electron acceptability of oxygen. ROS are generated as natural byproducts of energy metabolism and have crucial roles in cell signaling and homeostasis.

> 2. At the end of the introduction section, the author has to explain aim, objective and outcome of the present study.

Response:

Thank you for your suggestion. The aim has already been stated at the end of the Introduction as “Based on the abovementioned information, the present study aimed to apply a meta-analysis approach to shed light on the oxidative transcriptomes of insects based on publicly available data.” We have added the outcome of the study to the end of the Introduction as follows.

> This study provides a method, including the workflow developed, that can be a powerful tool for deciphering conserved networks in oxidative stress response.

> 3. What is the importance of Common Workflow Language? Why author had used this one in the present study? please justify.

Response:

Thank you for your suggestion. We have added detailed descriptions about the importance and merit of using CWL in the results section as follows.

> Using CWL, one can replicate the analysis without installation of software, regardless of the type and version of operation system used (MacOS, Linux, and Windows). CWL enables the use of the workflow very easily, without installing Trinity, which users often find difficult to install on their computers.

> 4. Table 3 should be uniform, there was an alignment issues by end of the table.

Response:

Thank you for your suggestion. We have made changes to ensure that there are no longer any alignment issues in Table 3.

> 5. In Table 4, Please give an expansion of "+" and “-"

Response:

Thank you for your suggestion. We have revised the title of Table 4 to include the meaning of “+” and “-“ as follows.

Table 4Drosophila melanogaster oxidative stress-related genes with ON-score feature (positive value (+) or not (-)) and Caenorhabditis elegans ortholog (yes or no)